# LEARNING ARTICULATED RIGID BODY DYNAMICS SIMULATIONS FROM VIDEO

**Eric Heiden**
University of Southern California
heiden@usc.edu

**Ziang Liu**
Stanford University

**Vibhav Vineet**
Microsoft Research

**Erwin Coumans**
NVIDIA

**Gaurav S. Sukhatme**
University of Southern California

## ABSTRACT

Being able to reproduce physical phenomena, ranging from light interaction to contact mechanics, simulators are becoming increasingly useful to more and more application domains where real-world interaction or labeled data is difficult to obtain. Despite the gain in attention, it requires significant human effort to configure simulators to accurately reproduce real-world behaviors. We introduce a pipeline that combines inverse rendering with differentiable simulation to create digital twins of real-world articulated mechanisms from depth or RGB videos. Our approach automatically discovers joint types and estimates their kinematic parameters, while the dynamic properties of the overall mechanism are tuned to attain physically accurate simulations. On a real-world coupled pendulum system observed through RGB video, we correctly determine its articulation and simulation parameters, such that its motion can be reproduced accurately in a physics engine. Having learned a simulator from depth video, we demonstrate on a simulated cartpole that a model-predictive controller can leverage such dynamics model to control nonlinear systems.

We provide further results and details on our project website at
https://eric-heiden.github.io/video2sim.

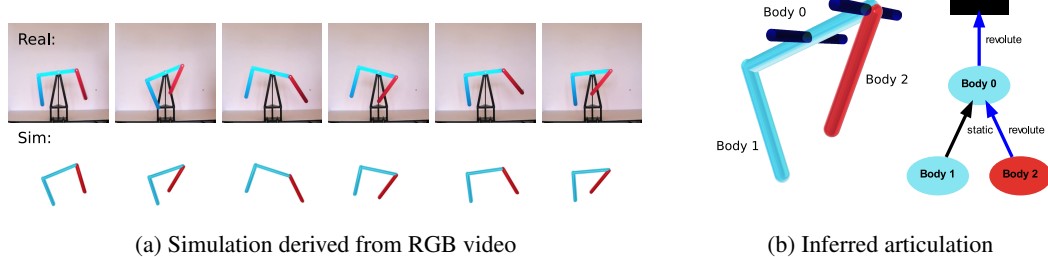

(a) Simulation derived from RGB video            (b) Inferred articulation

Figure 1: Our proposed framework infers articulated rigid body dynamics simulations from video. In this example, Rott's pendulum is identified from real RGB camera footage of the system in motion. In (a), the original video is compared to the simulated motion in the learned physics engine. In (b), the inferred joints are visualized on the left, where blue cylinders correspond to the axes of the revolute joints. The estimated kinematic tree is shown on the right.

# 1 INTRODUCTION

Simulators are one of the most capable world representations that can reproduce a vast range of behaviors in great detail through a variety of dynamics models. Provided their implemented models are calibrated correctly, these tools can generalize exceptionally well compared to most purely

data-driven models. They are an indispensable tool in the design of machines where the cost of prototyping hardware makes iterating in the real world prohibitively expensive. Robot control pipelines are often trained and developed in simulation due to the orders of magnitudes of speed-ups one can achieve by simulating a great variety of interaction scenarios in parallel faster than real time without causing damage in the early phases of training.

Nonetheless, it remains a challenge to leverage such tools for real-world tasks. Not only is there a sim2real gap due to the inherently incomplete model of reality these simulators implement, but it is often difficult to find the correct simulation settings that yield the most accurate results. Despite recent advances in bridging the sim2real gap (see Höfer et al. (2020) for an overview), deriving a Unified Robot Description Format (URDF) file or analogous scene specifications can pose a tremendous challenge when a real-world system needs to be simulated accurately.

In this work, we tackle the problem of automatically finding the correct simulation description for real-world articulated mechanisms. Given a depth or RGB video of an articulated mechanism undergoing motion, our pipeline determines the kinematic topology of the system, i.e. the types of joints connecting the rigid bodies and their kinematic properties, as well as the dynamical system properties that explain the observed physical behavior. Relying on camera input as observation signal to our pipeline opens the avenue to future work integrating simulators into embodied agents that can leverage their predictive power to reason about the physical world around them and make high-level decisions that leverage the semantic information that the simulator encodes.

## 2 APPROACH

As shown in Fig. 2, our proposed pipeline consists of four steps.

First, we find the instances of known objects in the first frame of the input video, as well as their segmentation maps via the Detectron2 (Wu et al., 2019) model. We train this instance segmentation network using our own two synthetic datasets consisting of (1) depth, and (2) RGB images of primitive shapes, such as capsules, spheres, and boxes, in different configurations and sizes observed from varying camera perspectives.

Next, we instantiate the 3D meshes for these shapes that have been identified in the input image in a differentiable rasterizer. We assume that such meshes, as well as the pose of the camera, are available; and leave the geometric shape and camera pose inference open for future work. As rasterizer we use nvdiffrast (Laine et al., 2020), a fast GPU-powered renderer that computes gradients of the 3D geometry w.r.t. to the pixel output. It supports both depth and RGB rendering, allowing us to infer the 3D poses of the object meshes via gradient-based optimization. The pose estimation is set up to minimize the L2 norm between the rendered and ground-truth image, where the rigid poses of the meshes are optimized. To improve convergence, we initialize the positions of the meshes via the centroids of the segmentation maps, and use parallel random restarts. We perform pose estimation for all the remaining frames in the input video, where the pose estimation of the rigid objects is initialized from the solution of the previous frame.

Given the pose trajectories of the rigid objects in the system, we next determine the topology of the mechanism, i.e. how the bodies are connected to each other by which type of joint. We follow a RANSAC approach (Fischler

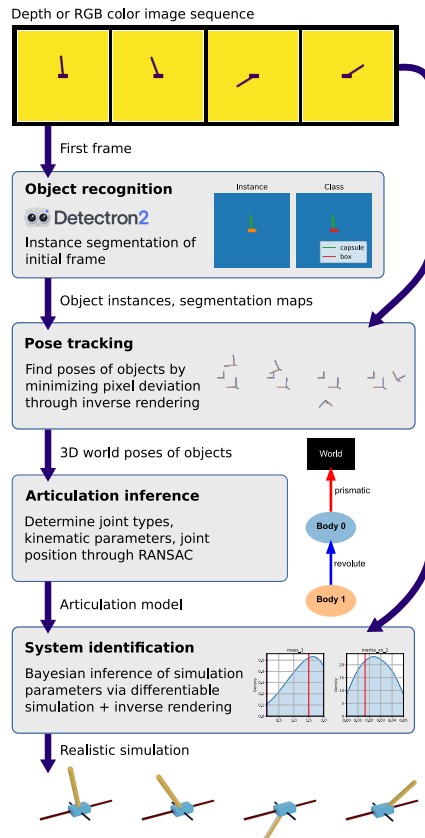

Figure 2: Pipeline of our proposed simulation inference approach that derives an articulated rigid body simulation from pixel inputs. The shown exemplary results generated by the phases in this diagram stem from the cartpole inference experiment from Sec. 3.1.

& Bolles, 1981) that yields robust results despite noisy pose estimates. For each pair of rigid bodies, we run the RANSAC algorithm on three different joint candidates: revolute, prismatic, and fixed joints. Further details on this inference process are given in Appendix A. These types of joints correspond to the most commonly encountered articulations in everyday objects. Given the joint parameters and model error for each of the joint candidates between bodies $i$ and $j$, we populate a cost matrix $C$ at indices $i$ and $j$ with the lowest cost, and memorize the corresponding joint parameters. Next, we determine the minimum spanning trees over $C$ which correspond to the distinct articulations in the scene. The root nodes of such articulations are processed in the final step where we determine how they are connected to the world frame, i.e. by one of the aforementioned joint types, or not connected at all, in which case we treat the articulation as free-floating. This process allows us to infer multiple articulated systems in the same video. Leveraging RANSAC, the joint inference is robust enough to tolerate imperfect pose estimates or occasional tracking outliers.

The articulation information is crucial in setting up the physics engine in the final phase of our pipeline. It determines the joint topology of the mechanism which we can now instantiate in a differentiable simulator, allowing us to learn the dynamical properties of the real system end-to-end from pixel observations through the differentiable rasterizer. In essence, the articulation inference phase allows us to find the non-differentiable structure of the mechanism, which we will now use to optimize all continuous variables in the simulator. While this means that our pipeline will fail in the following steps if we do not find the right kinematic structure (link connections and joint types) of the system, we can still recover from errors when kinematic properties, such as joint axes, could not be identified correctly. We use the Tiny Differentiable Simulator (Heiden et al., 2021b) that implements articulated rigid body dynamics and contact models, and calculates gradients for the dynamic and kinematic parameters of the mechanism. By attaching the previously defined meshes to the links in the mechanism, we can couple the physics engine with the differentiable rasterizer. Given the dynamical and kinematic system parameters, the physics engine produces a trajectory of joint positions. These generalized coordinates are translated to the 3D poses of the meshes via (differentiable) forward kinematics, so that the rasterizer can produce an image sequence. This simulated video is again compared against the real-world observations (L2 distance), allowing us to formulate an optimization problem that we can minimize through gradient-based optimization. Furthermore, we can leverage modern Bayesian inference algorithms, such as Stein Variational Gradient Descent (SVGD) (Liu & Wang, 2016), and a recently introduced constrained variant (CSVGD (Heiden et al., 2021a)) specifically tailored to trajectory-based inference problems. Those likelihood-driven methods allow us to efficiently find posterior distributions over simulation parameters to given the noisy observations from the real world, while leveraging gradient information of the simulator. We provide further details in Appendix B.

## 3 EXPERIMENTS

### 3.1 SIMULATED CARTPOLE

In our first experiment we consider a simulated cartpole, a nonlinear system where a rotating pole is attached to a cart that can move sideways. Given a depth image sequence of a duration of 2 s from simulation where the cartpole is moving passively starting with the cart at the center and the pole at an angle of 0.1 rad from its upright position, we set up our pipeline to learn a simulator for this mechanism. The results from each step of our pipeline are shown in Fig. 2. In the final parameter estimation phase, we infer the inertial properties of the rigid bodies (mass, center of mass, and the diagonal of the $3 \times 3$ inertial matrix) via the Bayesian inference algorithm CSVGD. As shown in the posterior plot for the two masses in Fig. 3, the predicted parameter distribution closely approximates the ground-truth parameters.

Having learned an accurate simulator, we now investigate its application in model-based control. We leverage Model Predictive Path Integral (MPPI) Williams et al. (2017), an information-theoretic model predictive control (MPC) algorithm that has been shown to

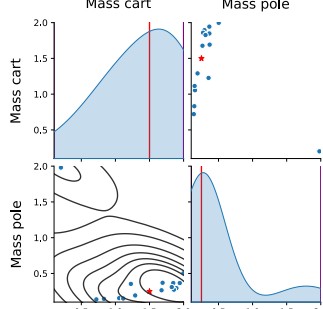

Figure 3: Parameter posterior distribution of the two link masses of the cartpole inferred from depth video via the multiple-shooting Bayesian inference approach CSVGD. The red lines and stars indicate the ground-truth parameters.

Reference system

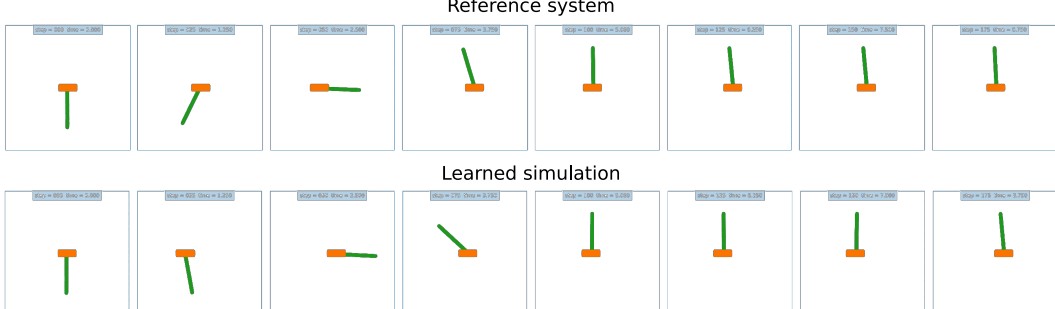

Learned simulation

Figure 4: Comparison of trajectories between a model-predictive controller leveraging the actual physics simulator (top row), and using the learned simulator as model (bottom row) to control the original cartpole system to swing up.

being particularly suited to control nonlinear systems. When tasked to swing up the cartpole from a configuration where the pole is pointing downwards, we observe that MPPI with our learned simulator as dynamics model achieves a performance close (within 86%) to the same controller equipped with the actual physics simulation that the cartpole is being evaluated in (see Fig. 4). On a cartpole balancing task (where the pole starts from an angle of $20°$ away from its upright position), the performance is approximately 95% of that of the reference controller.

## 3.2 Rott's Chaotic Pendulum

We now aim to identify a real-world chaotic mechanism – a coupled pendulum first analyzed by Nikolaus Rott (Rott, 1970). Rott's mechanism consists of two pendula: one L-shaped pendulum is attached to a fixed pivot via a revolute joint, and a single body is attached to this L-shaped pendulum via another revolute joint (see Fig. 1). Given a video taken with an RGB camera of such a mechanism, we aim to reconstruct a digital twin in simulation. Since we cannot rely on depth information to inform the 3D poses of the rigid bodies, we assume the mechanism to be a planar system. Therefore, the rigid body tracking system is constructed such that each body only has three degrees of freedom ($x$, $z$ position and yaw angle). Following our inference pipeline, we find the correct articulation shown in Fig. 1b, where the two revolute joints, as well as the static joint, and their parameters have been identified. Having optimized the simulation parameters (the inertial properties of the three links) via the Adam optimizer, we arrive at a simulation that closely reproduces the real camera footage (comparative snapshots of a 7 s video are shown in Fig. 1a).

## 4 Related Work

Articulation inference has been an important task in robotics, where interactive perception approaches have been proposed for a robot to determine how to interact with common household objects (Martin Martin & Brock, 2014; Hausman et al., 2015; Eppner et al., 2018). Such inference problem hinges on accurate pose estimates of the rigid bodies, which is why many early works relied on fiducial markers to accurately track objects and subsequently determine the articulations (Sturm et al., 2011; Niekum et al., 2015; Liu et al., 2019).

Learning-based approaches, such as ScrewNet (Jain et al., 2020) and DUST-net (Jain et al., 2021) infer single articulations from depth images, whereas our approach recovers multiple articulations between an arbitrary number of rigid objects in the scene. In Mu et al. (2021) and Noguchi et al. (2021) signed distance fields are learned in tandem with the articulation of objects to infer kinematic 3D geometry, without considering the dynamics of the system.

Entirely data-driven physics models often leverage graph neural networks to learn dynamical constraints between particles or bodies ((Battaglia et al., 2016; Xu et al., 2019; He et al., 2019; Sanchez-Gonzalez et al., 2020)).

Closer to our work is VRDP (Ding et al., 2021), a pipeline that similarly leverages differentiable simulation to learn the parameters underlying rigid body dynamics, but does not consider articulated systems. GradSim (Murthy et al., 2021) combines differentiable simulation with differentiable

rasterization, but requires the articulation to be known. Vid2Param (Asenov et al., 2019) adapts a variational recurrent neural network to predict physical parameters directly from videos, and has been applied to single rigid-body dynamics problems.

## 5 CONCLUSION

Our proposed pipeline allows the automatic inference of articulated rigid-body simulations from video by leveraging differentiable physics simulation and rendering. Our results on a simulated system demonstrate that we can achieve accurate trajectory predictions that benefit model-based control, while the learned parameters are physically meaningful. On a real-world coupled pendulum system, our approach predicts the correct joint topology and results in a simulation that accurately reproduces the real RGB video of the mechanism.

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

## A ROBUST ARTICULATION INFERENCE

Similar to Martin Martin & Brock (2014), given the sequence of world transforms $T_0^i$ of all rigid bodies $i$ in the scene, we determine the relative transform $T_i^j[t]$ for each unique pair of rigid bodies $i$ and $j$ at each time step $t$. We denote the rotation of a transform as $T.r$, which uses the 3D axis-angle representation. The translation of a transform is referred to as $T.p$.

**Revolute joint model** Determine joint axis $\mathbf{s}$, pivot point $\mathbf{p}$ and joint angle $q$ from two consecutive transforms at times $t$ and $t+1$:

$$\Delta r = T_i^j[t+1].r - T_i^j[t].r$$
$$\Delta p = T_i^j[t+1].p - T_i^j[t].p$$
$$\mathbf{s} = \frac{\Delta r}{\|\Delta r\|}$$
$$\mathbf{p} = T_i^j[t].p + \frac{\Delta r \times \Delta p}{\|\Delta r\|^2}$$
$$q = \|\Delta r\|$$

**Prismatic joint model** Determine joint axis $\mathbf{s}$ and joint position $q$:

$$\Delta p = T_i^j[t+1].p - T_i^j[t].p$$
$$\mathbf{s} = \frac{\Delta p}{\|\Delta p\|}$$
$$q = \mathbf{s} \cdot T_i^j[t+1].p$$

**Static joint model** The static joint is parameterized by the fixed relative transform $T_i^j$ between bodies $i$ and $j$.

Algorithm 1 summarizes our inference approach to determine the articulations between rigid bodies given their observed 3D motions. We first find the most likely joint types and corresponding joint parameters between unique pairs of rigid bodies via RANSAC for the three different joint models of revolute, prismatic and static joints. If no joint model could be found that matches the relative transform sequence between two rigid bodies, they are considered to be disconnected.

Having computed the cost matrix $C$ of the joint model errors from the previous RANSAC estimation, we find the minimum spanning forest via Prim's algorithm that we run on each component of the undirected graph described by the weighted adjacency matrix $C$. We select the root node $i$ from each minimum spanning tree as the top-level body in the kinematic tree to which all the rigid bodies within the same component are connected via the previously found joint models according to the hierarchy of the spanning tree.

Given the root node's time sequence of world transforms $T_0^i$, we determine the most likely joint model for the base of the articulated system again via RANSAC. If such a model has been found, the corresponding articulated mechanism is considered fixed-base and gets connected through this joint to the world. If no such joint model could be found, the articulation is floating-base and needs to be considered as such in the simulator (either by adding degrees of freedom corresponding to a rigid-body motion, or via a flag that ensures the mechanism is simulated as a floating-base system).

## B INFERENCE OF SIMULATION PARAMETERS

Leveraging Bayesian inference, we infer the dynamical parameters (which may include masses, link lengths, friction coefficients, etc.) of the mechanism. We model the inference problem as a Hidden Markov Model (HMM) where the observation sequence $\mathcal{X} = [\mathbf{x}_1, \ldots, \mathbf{x}_T]$ of $T$ video frames is derived from latent states $\mathbf{s}_t$ ($t \in [1..T]$). We assume the observation model is a deterministic function which is realized by the differentiable rasterization engine that turns a system state $\mathbf{s}_t$ into an observation image $\mathbf{x}_t$. The states are advanced through the dynamics model which we assume is

---

**Algorithm 1** Determine articulation of observed motion

---

**Input:** world transform sequence $T_0^i$ for each rigid body $i = 1..n$
$C = \mathbf{0}_{n \times n}$
**for** body $i \in \{1..n\}$ **do**
    **for** body $j \in \{i+1..n\}$ **do**
        Calculate relative transform sequence $T_i^j$
        Determine joint parameters $\theta_{joint}$ for revolute (r), prismatic (p), static (s) joint type given
$T_i^j$ via RANSAC, as well as their respective model errors $c_r, c_p, c_s$
        **if** RANSAC found at least one joint candidate **then**
            $C[i,j] = C[j,i] = \min\{c_r, c_p, c_s\}$
            Memorize $\theta_{joint}^*$ of candidate with lowest cost
        **else**
            $C[i,j] = C[j,i] = \infty$
        **end if**
    **end for**
**end for**
Determine minimum spanning forest on $C$, retrieve root bodies $I_{root}$
**for** body $i \in I_{root}$ **do**
    Construct kinematic tree $A$ rooted at body $i$, with parent-child connections from the corresponding minimum spanning tree and respective memorized joint parameters
    Determine joint candidate $\theta_{joint}$ via RANSAC given $T_0^i$
    **if** RANSAC found at least one joint candidate **then**
        Attach $A$ to world via lowest-cost joint model $\theta_{joint}^*$
    **else**                                                  ▷ floating-base case
        Attach $A$ to world via free joint
    **end if**
**end for**
**return** world model consisting of articulations

---

fully dependent on the previous state and the simulation parameter vector $\theta$. In our model, this transition function is the differentiable simulator that implements the articulated rigid-body dynamics equations and contact models. We use the Tiny Differentiable Simulator (Heiden et al., 2021b) that implements end-to-end differentiable contact models and articulated rigid-body dynamics following Featherstone's formulation (Featherstone, 2007).

Following Bayes' law, the posterior $p(\theta|\mathcal{X})$ over simulation parameters $\theta \in \mathbb{R}^M$ is calculated via

$$p(\theta|D_{\mathcal{X}}) \propto p(D_{\mathcal{X}}|\theta)p(\theta).$$

We leverage the recently introduced Constrained Stein Variational Gradient Descent (CSVGD) algorithm (Heiden et al., 2021a) that introduces constraints to the gradient-based, nonparametric Bayesian inference method SVGD (Liu & Wang, 2016). The constraint handling allows us to enforce parameter limits and optimize simulation parameters via multiple shooting. This technique splits up the trajectory into shooting windows for which the start states need to be learned. Defect constraints are introduced that enforce continuity at the start and end states of adjacent shooting windows. Despite requiring extra variables to be optimized, multiple shooting significantly improves the convergence of gradient-based parameter inference approach when parameters need to be inferred from long time horizons. Further details can be found in Heiden et al. (2021a).

