# OpenReview forum: "Learning Articulated Rigid Body Dynamics Simulations From Video"
_ICLR.cc/2022/Workshop/OSC — ICLR2022 OSC  Poster_

### Official Review · Reviewer_K4NC · 2022-03-15

**Rating:** 2
**Confidence:** 2

**Review:**

Given a video of a real-world system, the aim of this paper is to set up a corresponding system in simulation. To that end, the paper proposes a combination of methods that detect individual objects and infer their physical parameters and mutual joint types. The inference of articulation as a tree of bodies and joints is the main contribution.

Strengths:
* The proposed pipeline can correctly infer the individual bodies and joints of a real-world pendulum system with three bodies and three joints. This is done given only a video and prior knowledge of what objects we can encounter.
* While the method assumes access to the geometries of individual objects, the actual object recognition and pose detection from images is done without taking any shortcuts.
* The method is shown to enable control of a simulated cartpole.

Weaknesses:
* The evaluation of the individual components of the proposed system is lacking. It is unclear what the limitations are, except for only modeling simple objects.

Overall, this paper fits the theme of the workshop well and it reports interesting results.

---

### Official Review · Reviewer_Z1xh · 2022-03-16

**Rating:** 2
**Confidence:** 2

**Review:**

The authors introduce a pipeline for inferring articulated rigid body dynamics simulations from video, and evaluate their method on the simulated cartpole and real-world double pendulum. The pipeline consists of four stages: object recognition, pose tracking, articulation inference, and system identification.

As far as I can see, the work is novel and relevant to the workshop.

Strengths:
- results on real world system
- in the simulated cartpole environment, applying MPPI on their learned simulator achieves performance close to the same controller in the actual simulator

Weakness:
- results for control on the real world double pendulum are not provided - this would strengthen the paper
- Using Detectron2 requires providing supervision on object instances
- it does not look like later stages in the pipeline can provide feedback (e.g. gradients) to earlier stages of the pipeline if the outputs of earlier stages of the pipeline are poor. This is one of the advantages of an end-to-end system that the current stage-wise system does not have.

---

### Decision · Program_Chairs · 2022-03-24

Accept (Poster)